# MIMOSA: Multimodal Concept-based Representations

## Abstract

In recent years, deep learning-based architectures have significantly improved multimodal representation. However, interpretability remains challenging with traditional attention and gradient-based methods, offering limited insights into decision-making processes. Concept-based explainability provides intrinsic model interpretability by mapping raw data to higher-level abstractions, yet it has only been applied to unimodal data. We present MIMOSA (MultIMOdal concept-based repreSentAtions), a unified multimodal model that integrates concept-based interpretability. Our research shows that exploiting a joint multimodal conceptual representation achieves comparable accuracy with multimodal black-box models, surpassing approaches based on unimodal concepts. This unified representation also prevents misclassification of concepts between modalities and improves concept interventions. Through a concept decoder, MIMOSA can extract concept visualizations for each modality. Experimental results obtained from three distinct multimodal datasets substantiate the efficacy of our approach, showcasing enhanced interpretability in multimodal models.

## 1 Introduction

In recent years, there have been significant advancements in the development of deep learning models capable of classifying, understanding, and generating information. Multimodal models are a step forward, as they enable the integration and generation of diverse data types such as text, images, graphs, and audio (Guo et al., 2019; Sleeman IV et al., 2022; Manzoor et al., 2023), but also clinical and molecular data such as proteomic and transcriptomic (Reel et al., 2021; Lovino et al., 2022). Transformer-based architectures, initially designed for processing sequential data, have proven to be especially important in improving the performance of multimodal representation learning (Xu et al., 2023). These models can effectively integrate diverse data types into a cohesive representation by capturing long-range dependencies and contextual relationships through mechanisms such as self-attention (Vaswani et al., 2017; Devlin et al., 2018; Dosovitskiy et al., 2020). By leveraging the information contained in each modality, they can achieve a more comprehensive understanding of a given input sample.

Most multimodal models proposed in the literature deliver high performance but often function as black-box systems, lacking interpretability. The interpretability of such models is crucial, as it enables the extraction of meaningful insights and ensures model reliability and fairness in decision-making processes (Joshi et al., 2021; Chefer et al., 2021a). Although the attention mechanism inherent in transformer models (Vaswani et al., 2017) has been suggested to provide some degree of interpretability (Abnar & Zuidema, 2020; Chefer et al., 2021b), this alone is insufficient for a comprehensive understanding of the decision process across modalities (Jain & Wallace, 2019). However, while attention mechanisms can reveal *where* a model is looking, they do not fully explain *what* a model is seeing in a given input. This is crucial for informing its decision-making process.

Concept-based explainability has emerged as a recent approach to unraveling *what* a model sees within a given input (Rudin, 2019; Fel et al., 2023; Poeta et al., 2023). Moving beyond post-hoc interpretability approaches (Kim et al., 2018; Ghorbani et al., 2019), concept-based models (Koh et al., 2020; Chen et al., 2020) offer intrinsically interpretable networks that translate raw input data into higher-level abstractions, such as class attributes, object-parts, or prototypes. Concept-based

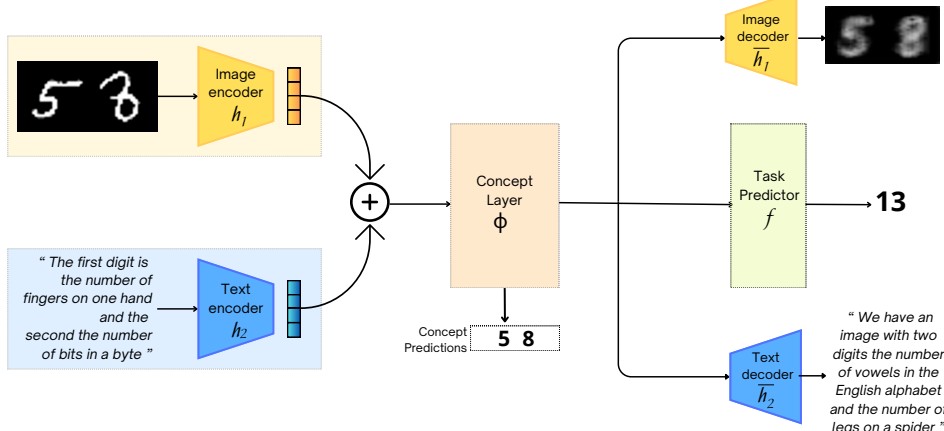

Figure 1: MIMOSA Architecture. On the left, encoders for the two modalities (image and text) are shown. Their representations are combined and passed to the concept layer. From the concept layer's output, we derive the prediction for the task (in this case, the sum of two digits), as well as visualizations of the concepts via two decoders, one for each modality.

models, however, have always been proposed for unimodal classification, such as image, graph, text, and tabular data (Ciravegna et al., 2023; Barbiero et al., 2023; Jain et al., 2022).

In this paper, we propose MIMOSA (MultIMOdal concept-based repreSentAtions), a novel unified multimodal model that integrates multimodal representation learning while integrating concept-based interpretability techniques. Currently, MIMOSA is focused on text and image modalities. A key feature is its ability to extract concept prototypes from the shared embedding space of these modalities, allowing for intuitive concept visualizations and enhancing model interpretability. Our contributions are as follows.

- *Accurate multimodal concept-based model.* Our model achieves accuracy greater or close to black-box models and higher on average than unimodal concept-based models.

- *Shared concept representation.* We employ a single concept representation shared across modalities avoiding discordant concept classification.

- *Independent concept decoding.* We extract concept visualization for each modality by attaching a concept decoder.

## 2 BACKGROUND

**Multimodal representations**   Multimodal representation emerged as a research field for creating machine learning models capable of jointly analyzing diverse data types (Ngiam et al., 2011). Initial approaches often relied on handcrafted features or shallow fusion methods. However, these methods had limited ability to capture complex inter-modal relationships. In recent years, the advent of deep neural networks enabled the fusion of several modalities (Baltrušaitis et al., 2018) to solve many tasks (Reed et al., 2022). This effort has been further propelled by transformer models with ad hoc pre-training, particularly for vision-language tasks (Radford et al., 2021; Zhai et al., 2022; Wang et al., 2021; Alayrac et al., 2022; Chen et al., 2022; Li et al., 2019). These models excel in various tasks such as image captioning, visual question answering, and cross-modal retrieval and can learn new tasks with very few training samples.

Different types of fusion strategies for the modalities have been proposed, including *late* strategies and *early* strategies (Gadzicki et al., 2020; Nagrani et al., 2021). These methods aim to create a unique latent space $z$, which can be either the result of $n$ different encoders $h_i$ or the result of a single encoder $h$ merging all input modalities $x_i, i = 1 \ldots n$. An illustration of the two fusion strategies are reported in Figure 2.

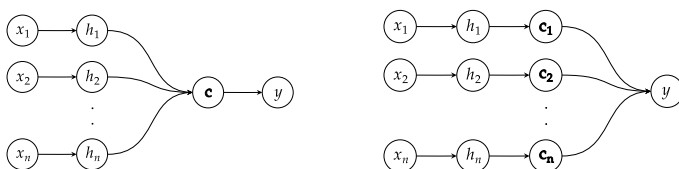

Figure 2: Illustration of early (left) vs. late (right) concept fusion strategies. On the left, fusion of the $n$ modalities occurs prior to generating the concept representations. On the right, concept representations for each of the $n$ modalities are obtained first and then merged just before the task prediction.

**Concept-based Models** Concept-based explainability has been proposed to enrich the explanations of standard XAI methods and incorporate human-understandable symbols (Poeta et al., 2023). This approach encompasses both post-hoc explainability methods (Kim et al., 2018; Ghorbani et al., 2019) and explainable-by-design models (Koh et al., 2020; Alvarez Melis & Jaakkola, 2018; Chen et al., 2020; Yang et al., 2023). Among the latter, concept-based models (Koh et al., 2020) explicitly create transparent deep neural networks by means of a dedicated layer (i.e., concept-bottleneck layer) representing intermediate attributes. The overall model can be described as $f \circ g : X \xrightarrow{g} C \xrightarrow{f} Y$, where $X \in \mathbb{R}^d$ represents the input space, $\hat{c} = g(x)$ is the concept encoder mapping the input to the concept space $c \in C \subset [0, 1]^l$ and $\hat{y} = f(g(x))$ is the task predictor mapping providing the final classification $y \in Y \subset [0, 1]^k$. This model not only improves the comprehension of the model decision but also permits interaction with it by means of concept interventions, i.e., modifications of the concept representations $\hat{c} := \bar{c}$ provided by a human expert with the aim of extracting counterfactual predictions $f(\hat{c}) \neq f(\bar{c})$ (Dominici et al., 2024).

The main issue of concept-based models lies in their limited generalization capability imposed by the concept-bottleneck layer. To overcome this, the Concept Embedding Model (CEM) (Espinosa Zarlenga et al., 2022) employs a sparse representation of the concepts. More in detail, in CEM the concept encoder represents the concepts as a tuple of concept score $\hat{c}$ and associated concept embeddings $\mathbf{c}$, i.e., $(\hat{c}, \mathbf{c}) = g(x)$, where $\mathbf{c} \in \mathbb{R}^{l,e}$, and $\mathbf{c_j}$ is an embedding of the $j$-th concept of dimension $e$. Each concept embedding $\mathbf{c_j}$ is conditioned to represent the associated concept by means of a shared concept predictor function: $\hat{c}_j = s(\mathbf{c_j})$[1]. Without relying on a constrained representation of the concepts, CEM task function $f(\mathbf{c})$ matches the generalization capability of end-to-end (E2E) black box models.

## 3 METHODOLOGY

### 3.1 MULTIMODAL CONCEPT REPRESENTATION

In this paper, we consider the case in which input samples are composed of $n$ modalities $x_i \in X_i \subset \mathbb{R}^{d_i}, i = 1, \ldots, n$ each of dimension $d_i$, and the task consists in the classification of the input samples into a single category $y$. We also require additional concept annotations $c$ to be available for the tasks at hand. As shown in Figure 1, MIMOSA models the overall problem as $f(g(x))$. This time, however, the concept encoder function $g(x)$ is composed of several modality-dependent encoders $h_i(x_i)$, which are aggregated together and further process to provide concept predictions $\hat{c}$ and concept embeddings $\mathbf{c}$ as:

$$\hat{c}, \mathbf{c} = g(x) = \phi \left( \bigoplus_{i=1,\ldots,n} h_i(x_i) \right), \tag{1}$$

where $\bigoplus$ represents a pooling operator (e.g., the mean, the max, or the sum) mapping the outputs of each encoder function $h_i$ into a single representation $\bigoplus : \mathbb{R}^{b,n} \to \mathbb{R}^b$, and $\phi$ is the neural module actually producing the concept embedding. In this paper, we considered the case where $\phi$ is modeled

---

[1]Actually, The concept embeddings are represented by a weighted sum of the positive and negative concept embeddings according to the concept prediction $\hat{c}_j$.

as a simple sum operator, but other operators can be used (also parametrized, e.g., a self-attention module). Similarly to CEM, $\mathbf{c}$ is composed of several embeddings, each representative of a single concept, $\mathbf{c} = [\mathbf{c_1}, \mathbf{c_2}, \ldots, \mathbf{c_l}]$ and it is fed to the task predictor $f(\mathbf{c})$ to provide the final prediction $\hat{y}$. Unlike CEM, however, $\mathbf{c}$ here represents the presence of the concepts across the modalities.

**Early vs late concept fusion**  As for standard multimodal models, the fusion process may occur at different stages. In the context of MIMOSA, a key difference is whether the fusion occurs before representing the concepts (as described above) or after, as proposed in Dominici et al. (2023) for unsupervised concept bottleneck models. In this latter case, concepts are represented and predicted for each modality, i.e., $\hat{c}_{i,j}$. As we will see in Section 4, however, this strategy presents two critical issues: (i) the interpretability of the model is lower because concept predictions across modalities may be discordant, i.e., the model may predict the presence of a concept in a modality but not in the others; (ii) due to the replication of the concept representations, concept accuracy may be lower.

### 3.2 Extracting Concept Prototypes

Even though the interpretability of concept-based models is higher than black-box models, what the same concepts represent is sometimes unclear. For this reason, a few methods were proposed to visualize concepts by means of standard XAI techniques, such as saliency maps (Li et al., 2018; Chen et al., 2019; Bontempelli et al., 2023). In this paper, rather than analyzing input representation in a post-hoc way, we propose to decode concepts explicitly. We propose to employ a set of decoders $\hat{x}_i = \psi_i(\mathbf{c})$ working on the concept embeddings and trained to reconstruct the input as follows: $\mathcal{L}_{dec}(x_i, \psi(\mathbf{c})) = |x_i - \psi_i(\mathbf{c})|$. In the experiments, for the image decoders we tested different distance functions, Mean Squared Error (MSE), $L_1$, and Structured Similarity SSIM, while for the textual ones we compute the Cross-entropy over the predicted words.

This approach is inspired by unsupervised and hybrid concept representations (Alvarez Melis & Jaakkola, 2018; Sarkar et al., 2022; Marconato et al., 2022) that reconstruct input samples with the aim of extracting an unsupervised disentangled concept representation or complete the set of supervised concepts. Instead, this work reconstructs the input to visualize which concept prototypes the network has learned.

## 4 Experiments

All experiments in this study are conducted using Python 3 and PyTorch (Paszke et al., 2019) and executed on a server equipped with 4 A6000 GPUs for computational efficiency. Implementing the concept embedding layer of MIMOSA is done using the `pytorch-explain` library (Barbiero, 2021). For each model and dataset, we performed three runs and reported mean and standard deviation. Further insights into the architectures and hyperparameters utilized in our experiments are available in our repository `https://anonymous.4open.science/r/mimosa`.

**Datasets**  We evaluate our model on three datasets: (1) `MNIST+` (Manhaeve et al., 2018) is a modified version of the renowned MNIST dataset (LeCun & Cortes, 2010). In this adaptation, each sample contains two handwritten digits, and the objective is to predict the sum of these paired digits. Each sample is labeled with concepts that correspond to the individual digits and is supplemented with a descriptive caption that articulates the content of the image. For instance, in Figure 1, the caption is "The first digit is [digit1], and the second is [digit2]". Notably, digit1 and digit2 are textual descriptions of each digit-concept (e.g., 5: the number of fingers in one hand) which have been randomly sampled among a list of 10 digit descriptions and inserted in one of 10 different templates describing the presence of two digit-concepts. More examples of caption templates for MNIST+ are given in the Appendix A.1. (2) The `cdSprites+` dataset Sejnova et al. is designed for benchmarking multimodal variational autoencoders. Comprising samples of size 64x64x3, this dataset is divided into levels, with each level incrementally increasing the image complexity and characteristics. Each sample is accompanied by a caption and a predefined set of attributes that serve as concepts. These attributes include 3 shape primitives (heart, square, ellipse), 2 sizes (big, small), 5 colors, 4 locations (top/bottom + left/right), and 2 backgrounds (dark/light), resulting in a total of 240 unique feature combinations. (3) `CUB` (Wah et al., 2011), a classification of bird species enriched by a comprehensive set of 112 bird features selected in Koh et al. (2020). Due to

Table 1: Task accuracy comparison. Best results per dataset are in bold; best per modality are underlined.

| Data | Model | MNIST+ | cdSprites+ | CUB |
|------|-------|--------|-----------|-----|
| IMG | CBM-Linear (Koh et al., 2020) | $0.3642 \pm 0.0046$ | $0.8067 \pm 0.0015$ | $0.5205 \pm 0.0263$ |
| | CBM-MLP (Koh et al., 2020) | $0.9038 \pm 0.0017$ | $0.8098 \pm 0.0004$ | $0.4062 \pm 0.0747$ |
| | CEM (Espinosa Zarlenga et al., 2022) | $\underline{0.9042} \pm 0.0022$ | $0.8117 \pm 0.0014$ | $\underline{0.6018} \pm 0.0129$ |
| | E2E | $0.9006 \pm 0.0035$ | $\underline{0.8144} \pm 0.0009$ | $0.5384 \pm 0.0931$ |
| TXT | CBM-Linear (Tan et al., 2024) | $0.3573 \pm 0.0018$ | $0.9242 \pm 0.0002$ | $0.1063 \pm 0.0266$ |
| | CBM-MLP (Tan et al., 2024) | $0.9027 \pm 0.0012$ | $0.9234 \pm 0.0008$ | $0.0851 \pm 0.0229$ |
| | CEM (De Santis et al., 2024) | $\underline{0.9110} \pm 0.0003$ | $\underline{0.9243} \pm 0.0006$ | $0.1909 \pm 0.0325$ |
| | E2E | $0.9092 \pm 0.0003$ | $0.9236 \pm 0.0006$ | $\underline{0.2393} \pm 0.0287$ |
| IMG + TXT | CBM-Linear | $0.3847 \pm 0.0113$ | $0.9832 \pm 0.0003$ | $0.4423 \pm 0.0579$ |
| | CBM-MLP | $0.9842 \pm 0.0016$ | $0.9854 \pm 0.0001$ | $0.3683 \pm 0.0999$ |
| | MIMOSA (Ours) | $0.9912 \pm 0.0016$ | $0.9861 \pm 0.0002$ | $0.6141 \pm 0.0600$ |
| | SHARCS (Dominici et al., 2023) | $\mathbf{0.9978} \pm \mathbf{0.0002}$ | $\mathbf{0.9872} \pm \mathbf{0.0003}$ | $0.4553 \pm 0.0581$ |
| | E2E | $0.9758 \pm 0.0038$ | $0.9861 \pm 0.0003$ | $\mathbf{0.7552} \pm \mathbf{0.1061}$ |

the multimodal nature of our inputs, we also use an extended version of the CUB dataset introduced by Reed et al. (2016), which incorporates descriptive captions alongside bird images. To ensure alignment between the captions and the corresponding images, we conduct a careful process of refining the dataset. As a result, we have an improved version of the CUB dataset, which includes meticulously aligned images, selected concepts, and captions.

**Architectures** For all three datasets, we utilize a ResNet50 (He et al., 2016) pretrained on ImageNet (Deng et al., 2009) as the image encoder. For the text encoder, we employ the BERT base uncased (Devlin et al., 2018), hereafter referred to as BERT. For MNIST+ and cdSprites datasets, we use the *base* version, while for CUB, we use the *large* version, as the complexity of the fine-grained bird classification task benefits from a larger model capacity. The representations from both the image and text encoders are passed through a linear layer to map their sizes to 512 dimensions, after which they are summed.

The concept embeddings module (CEM) is implemented as described in Espinosa Zarlenga et al. (2022) with different embedding sizes depending on the dataset. For MNIST+ and cdSprites+, the embedding size is set to 16, and for CUB 64. These concept embeddings are then input to a task predictor consisting of a sequence of linear layers. Additionally, the concept embeddings are used for concept visualization via decoders tailored for both modalities. For MNIST+ and cdSprites+ text decoding, we use a GPT-2 model (Radford et al., 2019) while, for CUB we used a T5 model (Raffel et al., 2020). A convolutional decoder is employed for image decoding for the MNIST Addition and cSprites+ datasets. Given the higher complexity of the CUB dataset images, we utilize a Stable Diffusion model (Rombach et al. (2022)) as an image decoder for this dataset.

**Compared Models** We evaluate the performance of MIMOSA against unimodal and multi-modal approaches. For the unimodal models, we consider two concept-based models, i.e., Concept Bottleneck Models (CBM) (Koh et al., 2020) and CEM (Espinosa Zarlenga et al., 2022), and a black-box end-to-end model. The latter only comprises the encoders and the task predictor, without the concept layer. For CBM, we consider two settings: one using a linear layer and the other using a multi-layer perceptron (MLP). Although CBM and CEM were initially designed for image data, they have been recently extended to the textual fields, respectively in Tan et al. (2024) Furthermore, we generalize CBM to multimodal inputs by applying early fusion of both modalities—image and text—as previously explained for MIMOSA, which itself generalizes CEM by performing early fusion of these inputs.

So, in the multimodal setting, we evaluate MIMOSA, the generalized version of CBM (in both its variants), an end-to-end multimodal model, and SHARCS— the concept-based model proposed by Dominici et al. (2023). SHARCS extracts distinct conceptual representations for each modality, subsequently integrating them through a late fusion process. This approach stands in contrast to ours, as MIMOSA employs early fusion to integrate concepts from both modalities.

Table 2: Comparison of Concept Accuracy scores across different models. Unless otherwise noted, all standard deviations are below 0.0001. Best results per dataset are in bold; best per modality are underlined.

| Data | Model | MNIST+ | cdSprites+ | CUB |
|------|-------|--------|-----------|-----|
| IMG | CBM-Linear | 0.9895 | 0.9752 | $0.9159\pm_{0.0071}$ |
| | CBM-MLP | 0.9895 | 0.9753 | $0.8776\pm_{0.0212}$ |
| | CEM | $0.9896 \pm_{0.0002}$ | 0.9754 | $0.9122\pm_{0.0045}$ |
| TXT | CBM-Linear | 0.9868 | 0.9823 | $0.8312\pm_{0.0195}$ |
| | CBM-MLP | 0.9869 | 0.9823 | $0.8347\pm_{0.0102}$ |
| | CEM | $0.9872 \pm_{0.0002}$ | 0.9823 | $\underline{0.8505}\pm_{0.0013}$ |
| IMG + TXT | CBM-Linear | 0.9959 | 0.9989 | $0.8877\pm_{0.0087}$ |
| | CBM-MLP | 0.9957 | $\underline{\mathbf{0.9991}}$ | $0.8751\pm_{0.0256}$ |
| | MIMOSA (Ours) | $\mathbf{0.9960} \pm_{\mathbf{0.0002}}$ | 0.9990 | $\mathbf{0.9206}\pm_{\mathbf{0.0112}}$ |
| | SHARCS_IMG | $0.9896\pm_{0.0002}$ | $0.9753 \pm_{0.0002}$ | $0.4273\pm_{0.30340}$ |
| | SHARCS_TXT | $0.9870\pm_{0.0006}$ | 0.9823 | $0.5222\pm_{0.0623}$ |

## 4.1 MULTIMODAL CONCEPT-BASED REPRESENTATION ACCURACY

We evaluate the task and concept accuracy of MIMOSA against the unimodal and multimodal base-lines. Table 1 reports the task accuracy for the three evaluated datasets. First, we observe that the multimodal models consistently outperform their unimodal counterparts across all approaches, demonstrating the effectiveness of integrating multiple modalities to improve task performance. On the simpler, synthetic datasets MNIST+ and cdSprites+, SHARCS achieves the highest performance, with our MIMOSA model following closely behind. In these datasets, the concepts are simpler and less diverse, leading us to argue that the separate representation used by SHARCS does not introduce conflicting concept predictions, making it sufficient for accurate task predictions. Notably, the end-to-end (E2E) model performs slightly worse or on par with the concept-based models, suggesting that the concept information is not only sufficient but also aids in improving task accuracy.

For the more complex, real-world dataset CUB, as expected, the multimodal black-box E2E model outperforms all concept-based models. The E2E model directly learns the task prediction without relying on intermediate concept representations, which boosts performance but sacrifices inter-pretability. MIMOSA achieves the best performance among the concept-based models and offers interpretability. Specifically, MIMOSA achieves a significant accuracy improvement of +0.1588 over SHARCS, the runner-up. For this real-world dataset, the unified representation of concept embeddings proves beneficial to task performance. We will investigate the impact of the shared representation in Section 4.2.

Table 2 compares the concept accuracy among the various evaluated methods. Our model achieves the highest concept accuracy for MNIST+ and CUB, and ranks second for cdSprites+, with only a marginal difference of 0.0001 compared to CBM-MLP. Concept accuracy for SHARCS has two distinct values: one related to concepts derived from images and the other related to concepts derived from text. As shown in the table, SHARCS exhibits lower concept accuracy compared to MIMOSA, especially for the real-dataset CUB. Moreover, in this case, its concept accuracy from images has a high standard deviation (0.3). These outcomes suggest that the presence of duplicate representations for concepts, along with a separate conceptual space for images and text, may lead to the potential for discordant concept predictions, lower concept accuracy, and variability. The analysis of this concept discordance will be explored in the following section.

## 4.2 COMPARING SHARED AND SEPARATE EMBEDDINGS

MIMOSA performs an early fusion of the modalities at the early stages of the model, prior to the concept layer. We show that the early fusion and the consequent unified concept representation enhance concept accuracy by preventing conflicts or mismatches between concepts. A mismatch occurs when the concept representation of the modalities does not agree with the concepts present in the input. This may occur for late fusion modalities, as adopted in SHARCS, where concepts are

embedded independently in each modality, resulting in separate concept predictions. For instance, if the image modality identifies a square while the text modality does not, there is a mismatch between the concepts predicted from the image and those predicted from the text.

We experimentally evaluate how often mismatches occur between the predicted concepts from the two modalities in SHARCS. On the CUB dataset, 54.13% of the time, concepts are discordant between modes. On MNIST+ the concepts are discordant 2.44% of the time and 4.10% on cdSprites+, when working on clean samples. However, when injecting noise into data representations simulating an out-of-distribution scenario (similarly to Shin et al. (2022) for concept interventions), the discordancy also becomes important on the toy dataset with 26.49% on MNIST+ and 8.88% on cdSprites+. Also, take into consideration that we considered each concept separately, thus computing a discordancy only when the single concept was such. If we were considering concept predictions as concordant only when all concept predictions were concordant for a given sample, the results would have been worse.

The high number of mismatches affects both concept accuracy and interoperability. In terms of concept accuracy, as we observed in Table 2, SHARCS obtained a lower concept accuracy, which can be linked to the mismatches. In terms of interpretability, when concepts are discordant, it becomes difficult for practitioners to interpret task predictions in terms of concepts. In contrast, MIMOSA has, by design, none mismatches as it uses a unified concept representation. Hence, users can can directly interpret predictions clearly and directly through the model's concepts.

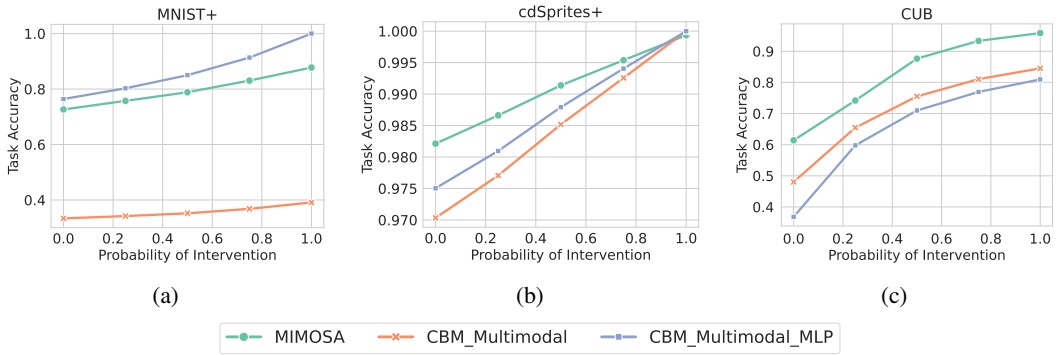

Figure 3: Concept intervention for MNIST+, cdSprites and CUB datasets.

### 4.3 EFFICACY OF INTERVENTION IN MULTIMODAL CONCEPT-BASED MODELS

Concept intervention involves identifying and modifying the internal representations of a model, i.e., the concept representation, to influence and potentially improve the model's behavior. To implement the concept intervention, we follow the methodology outlined in Espinosa Zarlenga et al. (2022), where we monitor task accuracy under varying levels of intervention. This process involves conditioning interventions on the probability of changing specific concepts, with higher probabilities indicating more substantial interventions. For instance, setting the probability of changing a particular concept to 0.75 means there is a 75% chance that this concept will be modified during the intervention process. We control the level of intervention, defined in terms of intervention probability, to observe how changes in concept embeddings impact overall model performance.

To evaluate the effectiveness of concept intervention on the MNIST+ and cdSprites+ datasets—both of which are relatively simple—we adopted the technique proposed in Shin et al. (2022). Specifically, during the interventions at test time, a Gaussian noise of unit mean and variance is added to the input representation of the sample $\phi(\bigoplus_i h_i(x_i))$. The idea is to simulate an out-of-distribution scenario where concept prediction accuracy necessarily decreases, and the support of an expert becomes important. The primary objective still remains to determine whether intervening on the concepts leads to an improvement in task accuracy, however, the technique allows for a more accurate evaluation of the intervention's impact since the baseline accuracy (with no intervention) for these two datasets would otherwise be too high to meaningfully observe improvements. Figure 3 shows the task accuracy after intervention on concepts varying the degree of intervention probability. Note

that when the probability is 0, we refer to the model with no intervention. For MNIST+, interventions on CBM Multimodal (Linear) and MIMOSA are more effective in improving task accuracy than those on CBM, whose accuracy in the clean scenario remains limited due to the non-linearity of the task at hand. CBM Multimodal MLP starts with lower accuracy, and the interventions allow the model to achieve up to 40% in accuracy. On cdSprites+, a high degree of intervention results instead in a similar improvement and final accuracy. The interventions on the CUB dataset have the most significant impact on MIMOSA compared to the other evaluated conceptual multimodal models.

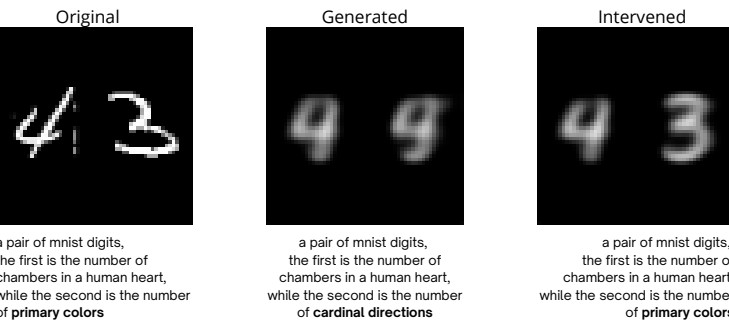

Figure 4: Example of concept intervention for MNIST+. This example illustrates how the application of targeted interventions on concepts triggers changes in both the visual representation and the accompanying textual description.

Figure 4 provides a visual and textual example of an intervention. This is made possible in MIMOSA through the employment of text and image decoders, which allow for the visualization of conceptual interventions on conceptual prototypes. The first sample is the original input, which represents a 4 and 3 and is wrongly predicted as having a sum of '8'. Practitioners may wonder why the model incorrectly made this prediction. Thanks to MIMOSA's ability to extract concept prototypes, we can visualize the predicted concepts, specifically '4' and '4', as reported in the second sample. Further details about the visualization are provided in the next section. In this sample, we observe '4' as the predicted digit instead of '3', and the generated text appropriately describes this concept as the "number of cardinal directions". Once practitioners identify the misclassification of concepts, they can intervene effectively. The final sample displays the generated prototypical image after the intervention, where we now visualize '4' and '3,' with the generated text accurately reflecting these concepts as the "number of primary colors".

## 4.4 QUALITATIVE ANALYSIS OF REPRESENTATIVE CONCEPT EXAMPLES

MIMOSA stands out from other methods by enhancing interpretability through the visualization of the concepts that influence the model's predictions. Users can gain valuable insights into how the model perceives the specific concepts it adopts.

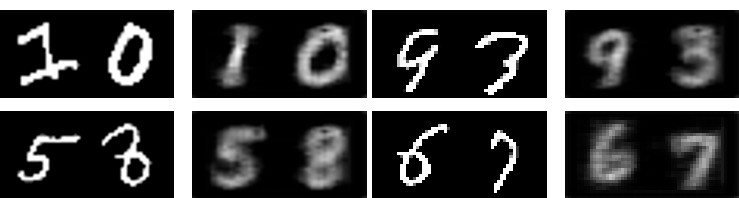

Figure 5: Examples generated by the image decoder. Notably, the visualization of the concepts obtained through the image decoder diverges from the original sample, as evidenced by the differences between the digit '1' in the initial and generated images. This suggests that the decoder is displaying what the model has internalized as the concept of '1', rather than directly reproducing the input sample.

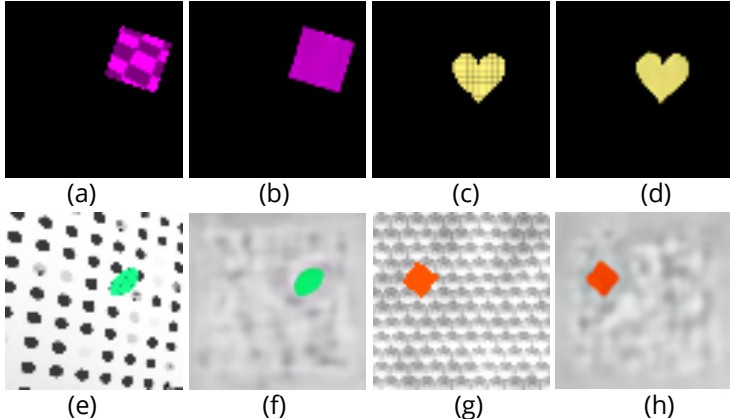

|  |  |  |  |
|---|---|---|---|
| (a) | (b) | (c) | (d) |
| (e) | (f) | (g) | (h) |

Figure 6: CdSprites input images (left) vs generated images (right) with MIMOSA. One can appreciate how the generated images represent the concept information the model has learned, as they don't contain spurious information (e.g., the shape, or background patterns)

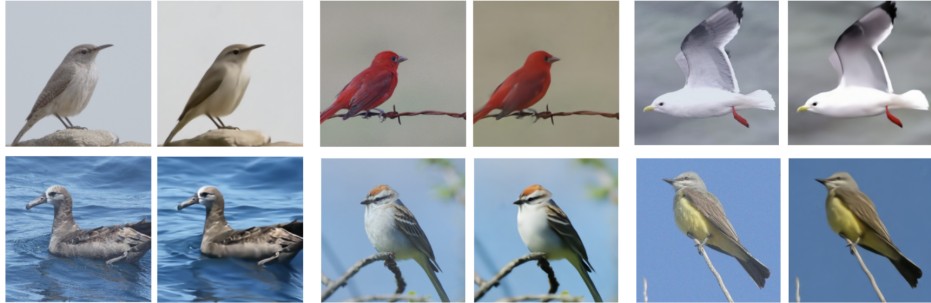

Figure 7: Examples of concept visualization for the CUB dataset. These are generated by means of the stable diffusion model, from the concept embeddings only.

As illustrated in Figure 1, MIMOSA not only provides concept predictions for a user to examine, but also allows the visualization of concept embedding by integrating an image decoder and a text decoder. This process results in reconstructed representations that reveal how the model interprets specific concepts. In the following, we provide examples focusing, for simplicity, on the generated prototypical images. For instance, in Figure 5, when processing the digit "1", the visualization shows a stylized version rather than a precise replication of the input instance. This reconstruction accentuates the salient features that the model associates with the concept of "1", not the input itself. The same principle applies to other digits, each offering a unique insight into the model's interpretation.

Figure 6 shows examples of generated images for the cdSprites+ datasets. Figure 6(a) shows an input image, representing a pink square with a grid pattern on a black background, while Figure 6(b) shows the generated image. Here, the prototypical image effectively represents all 4 concepts of this dataset: shape (square), size (big), color (pink), location (top right), and background (dark). Notably, the grid pattern of the square is, correctly, not reproduced as it is not a concept. Similar considerations apply to the other examples. Figure 7 reports examples for the CUB dataset. Here, the decoder successfully reconstructs from the concept embeddings, with a noticeable resemblance to the input image. While showing a strong generative performance of the models, these images also suggest potential information leakage in the concept representation (Havasi et al., 2022; Marconato et al., 2022), i.e., where additional information beyond concepts might be encoded. Nevertheless, concept embeddings retain relevant concept-based information, as we demonstrated by concept intervention experiments.

## 5 RELATED WORK

Multimodal Explainable AI targets explaining the behavior of multimodal models processing multiple types of input data simultaneously (Rodis et al., 2023). Most current solutions for multimodal explainability extend existing XAI, originally designed for unimodal models, to the multimodal setting. These methods typically provide local explanations by analyzing the behavior of individual predictions in a post-hoc manner, i.e., they attempt to explain the decisions of an already trained and otherwise opaque model. For example, DIME (Lyu et al., 2022) adapts the explanation method LIME (Ribeiro et al., 2016) to compute the contribution of the input of each modality and their multimodal interactions. Similarly, various approaches explain predictions in Visual Question Answering tasks by generalizing Integrated Gradient (Mudrakarta et al., 2018), Layerwise Relevance Propagation (Sun et al., 2020), or Guided-backpropagation (Nam et al., 2017). Other approaches leverage inherent model proprieties, such as the attention mechanism, to generate attention maps highlighting important features across one or more modalities (Lu et al., 2016). All these mentioned methods provide insights into *where* the model is focusing on by identifying salient parts of the input, such as regions of an image or significant tokens in text. However, these approaches fall short in determining the *what* the model is considering in its decision-making process.

The challenge of identifying "what" the model is considering for its predictions is a central goal of concept-based explainability (Poeta et al., 2023). Concept-based methods aim to explain model behavior using higher-level abstractions or concepts that are more aligned with human understanding. The work of Asokan et al. (2022) goes in this direction by extending Testing with Concept Activation Vectors (Kim et al., 2018) to a multimodal scenario, specifically for multimodal emotion recognition. However, this approach, like all the above mentioned, operates in a post-hoc manner, thus only approximating the behavior of a black box model.

In contrast, our approach aims to develop a transparent-by-design model that achieves high performance while simultaneously revealing the reasons behind its prediction through concepts. Concept-based models, by design, offer intrinsic transparency by translating raw input data into higher-level concepts, such as class attributes, object parts, or prototypes (Koh et al., 2020; Espinosa Zarlenga et al., 2022; Li et al., 2018; Chen et al., 2019). However, these models are limited to unimodal settings, focusing on a single modality like images, graphs, text, or tabular data.

Our proposed model addresses this gap by introducing a multimodal concept-based framework that operates in a multimodal setting over a shared and unified concept representation. As a result, we can explain the model prediction directly in terms of the interpretable concepts. Closely related to our approach is SHARCS (Dominici et al., 2023), which also proposes a multimodal concept-based framework. However, SHARCS extracts separate concept representations for each modality and combines them through late fusion. As our experiments demonstrate, this disjoint concept representation leads to lower concept accuracy and discordant and inconsistent concept predictions. Moreover, our approach includes a set of decoders, one for each modality, which facilitates the extraction of prototypes. This crucial component enables the visual representation of the learned concepts, offering deeper insight into the model's internalized understanding of the data.

## 6 CONCLUSION

MIMOSA is a novel multimodal concept-based approach integrating an early fusion of image and text data. This method enhances the model's ability to learn and represent shared concepts across different modalities, offering a more unified and interpretable framework. By incorporating modality-specific decoders for the extraction of prototypes, MIMOSA provides a visual representation of learned concepts, facilitating better interpretability and understanding of the model's decision-making processes. The early fusion strategy, combined with concept-based intervention, proves effective in improving task accuracy, especially in more complicated datasets like CUB. Overall, MIMOSA demonstrates its potential to advance concept-based explainability in multimodal contexts, setting the stage for further exploration and application.

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

# A  APPENDIX

## A.1  ADDITIONAL EXPERIMENTAL DETAILS

**Text templates for MNIST+ Captions**   In the MNIST+ datasets, a variety of templates were employed to generate diverse captions for the image pairs. These templates introduce flexibility and variability in describing the two digits. Each caption consists of two components: a fixed portion, referred to as the *text template*, and a variable portion, the *digits template*, which adapts to the specific digits displayed in the images.

Below is a list of the *text templates* used:

- "The first digit is X, and the second is Y."
- "A number is X, and the other is Y."
- "A picture with two numbers where one number is X, while the second number is Y."
- "There are two numbers in this image, the first digit is X, the second digit is Y."
- "Two numbers, the first one is X, the second one is Y."
- "An image with two digits, on the left X, on the right Y."
- "A pair of digits, X and Y."
- "We have an image with two digits, X and Y."
- "Two digits in this image, the left one is X, the right one is Y."
- "A pair of MNIST digits, the first is X, while the second is Y."

Below is a selection of templates used to describe the digit pairs in the MNIST+ datasets. These templates were generated through an interactive process with ChatGPT, where the model was instructed to create variations to describe each digit. Specifically, ChatGPT was tasked with providing 10 unique descriptions for each digit, resulting in a diverse set of expressions that offer more flexible ways to caption the figures.

Below is the list of some of the *digit templates* that were generated:

- **0:**
    - the all-round digit
    - the null element for addition
    - the only digit that represents nothingness
    - the placeholder that gives value to other digits
- **1:**

- – the multiplicative identity in arithmetic
- – the smallest positive integer
- – the number of moons Earth has
- – the lone digit that stands tall
- **2:**
  - – the smallest and first even prime number
  - – the number of wings on most birds
  - – the pair that makes a couple
  - – the smallest prime that divides evenly
- **3:**
  - – the first odd prime number greater than two
  - – the number of sides on the simplest polygon
  - – the digit often associated with luck and folklore
  - – the number of primary colors
- **4:**
  - – the smallest composite number
  - – the number of seasons in a year
  - – the number of cardinal directions
  - – the number of legs on most chairs
- **5:**
  - – the halfway mark between the first double digits
  - – the number of fingers on one hand
  - – the number of vowels in the English alphabet
  - – the number often associated with balance and harmony
- **6:**
  - – the smallest perfect number
  - – the number of faces on a standard cube
  - – the number of strings on a standard guitar
  - – the atomic number of carbon
- **7:**
  - – the number often considered lucky in many cultures
  - – the number of continents on Earth
  - – the number of colors in a rainbow
  - – the days in a week
- **8:**
  - – the cube of the smallest prime number
  - – the number of legs on a spider
  - – the number of vertices on an octagon
  - – the number of bits in a byte
- **9:**
  - – the highest single-digit number
  - – the number of planets in our solar system (if counting Pluto)
  - – the number of lives a cat is said to have
  - – the number of innings in a standard baseball game

