# OpenReview forum: "MIMOSA: Multimodal Concept-based representations"
_ICLR.cc/2025/Conference — ICLR 2025 Conference Withdrawn Submission_

### Official Review · Reviewer_pUyV · 2024-10-30

**Soundness:** 3
**Presentation:** 3
**Contribution:** 3
**Rating:** 3
**Confidence:** 4

**Summary:**

This paper introduces MIMOSA (MultIMOdal concept-based repreSentAtions), a novel multimodal framework that leverages concept bottleneck models (CBMs) to address interpretability challenges in multimodal tasks. By jointly learning shared concepts from image and text data, MIMOSA offers a unified representation that enhances transparency in multimodal decision-making, overcoming trade-offs inherent in traditional CBMs. Experimental results demonstrate that MIMOSA outperforms state-of-the-art methods on various benchmarks, providing a promising approach for interpretable multimodal learning.

**Strengths:**

1. Improved Multimodal CBM: The paper successfully extends Concept Bottleneck Models (CBMs) to the multimodal domain, addressing limitations and trade-offs inherent in traditional CBM approaches. MIMOSA demonstrates improved performance and a better balance of interpretability and accuracy.
2. Consistent Multimodal Representation: MIMOSA introduces a shared concept representation to ensure consistent representation of concepts across different modalities, enhancing transparency and interpretability in multimodal models.
3. Independent concept decoding: The model's independent concept decoding enables detailed visualization of modality-specific concepts, providing valuable insights into the decision-making process and boosting model trustworthiness.
4. Demonstrated Performance on Benchmark Datasets: The paper provides strong empirical evidence of MIMOSA's effectiveness through experiments on well-established benchmark datasets like MNIST++ and CUB.

**Weaknesses:**

1. Detailed Model Architecture: A more detailed explanation of the model architecture, including the specific structure of the function 𝑓 and decoder, would enhance the clarity of the technical implementation.
2. Figure 1 Alignment: In Figure 1, it is unclear where the network 𝜓 is located. Aligning Figure 1 with the main text for consistency and clarity would improve readability and understanding of the model components.
3. Inclusion of Recent References: The references section should include recent relevant papers such as (1) LaBO, (2) BotCL, and (3) ResCBM to provide a comprehensive review of the state-of-the-art.

   (1) Oikarinen, T., Das, S., Nguyen, L. M., & Weng, T. W. Label-free Concept Bottleneck Models. In The Eleventh International Conference on Learning Representations.

   (2) Wang, B., Li, L., Nakashima, Y., & Nagahara, H. (2023). Learning bottleneck concepts in image classification. In Proceedings of the ieee/cvf conference on computer vision and pattern recognition (pp. 10962-10971).

   (3) Shang, C., Zhou, S., Zhang, H., Ni, X., Yang, Y., & Wang, Y. (2024). Incremental residual concept bottleneck models. In Proceedings of the IEEE/CVF Conference on Computer Vision and Pattern Recognition (pp. 11030-11040).

4. Clear Novelty Distinction: A clearer delineation of MIMOSA's novelty, particularly in relation to recent label-free CBMs (LaBO) and ResCBM (as suggested in item 3) approaches, would strengthen the paper's contribution.
5. Limited Dataset Scope: The paper uses a limited number of datasets for the experiments. Expanding the experimental evaluation to include additional datasets like CUB, CelebA, CheXpert, and AWA2 would provide a more comprehensive performance assessment.
6. Text Data Source Clarification: Specifying the source of the text data used in the experiments would enhance the contextual understanding of the dataset.
7. Expanded Model Comparisons: The experimental section includes a limited number of comparative models, primarily CBM and CEM. To provide a more comprehensive evaluation, it is recommended to incorporate recent CBM models such as Post-hoc CBM, LaBO, and ECBM, as suggested in item 3.
8. Negative Intervention Experiments: Incorporating negative intervention experiments would provide a more complete assessment of MIMOSA's behavior under different intervention scenarios.

**Questions:**

See weaknesses

---

> ### Author Response · Authors · 2024-11-20
> **Response to reviewer pUyV (1/2)**
>
> Dear Reviewer pUyV,
>
> Thank you for your valuable feedback.
>
> Below, we provide responses to the specific concerns raised:
>
> ---
> #### 1. Detailed Model Architecture
> We appreciate your suggestion to provide further details about the model structure. Below is a breakdown of the *MIMOSA* architecture:
>
> - **Encoders:**
>   - *Image Encoder*: Utilizes a pre-trained ResNet50 backbone.
>   - *Text Encoder*: Initialized with a pre-trained language model (BERT base for MNIST+ and cdSprites+ datasets, BERT large for CUB).
>
> - **Concept Embedding Layer:**
>   - Based on the implementation from Espinosa Zarlenga et al. (2022) (CEMs).
>   - Trained from scratch using binary cross-entropy loss for concept predictions.
>
> - **Task predictor f:** A feedforward neural network with two linear layers separated by a LeakyReLU activation and Dropout for regularization. It maps concept embeddings to task predictions and is trained using cross-entropy loss.
>
> - **Decoders:**
>   - For MNIST+/cdSprites+:
>     - Image decoders are convolutional neural networks trained from scratch. Losses tested include Mean Squared Error (MSE), L1 distance, and Structural Similarity Index (SSIM).
>     - Text decoders compute cross-entropy loss over predicted captions.
>   - For CUB:
>     - Image decoder employs a pre-trained Stable Diffusion v1-4 model. Unlike conventional usage, where the model is prompted with text for inference, we fine-tune the decoder using the concept embeddings generated by MIMOSA. These embeddings act as a representation of concepts in a latent space. This fine-tuning allows Stable Diffusion to visualize the concepts directly, even without textual prompts. The fine-tuning process uses SSIM as the primary loss to ensure alignment between visualized concepts and the input data.
>
> These details will be included in the revised paper.
> #### 2. Training Objectives
> *MIMOSA* uses the following objectives:
>
> - **Concept Prediction Loss:** Binary Cross Entropy Loss for supervised concept predictions.
> - **Task Loss:** Cross-Entropy loss for the final classification task.
> - **Reconstruction Loss:** Modality-specific losses (e.g., MSE and SSIM for images, Cross-Entropy for text) ensure the alignment of shared concept embeddings with input modalities.
>
> The overall training objective combines these losses:
>
> $
> \text{L} = \lambda_\text{task} L_\text{task} + \lambda_\text{concept} L_\text{concept} + \lambda_\text{dec} L_\text{dec}
> $
>
> #### 3.  Figure 1 Alignment
>
> Thank you for noting the ambiguity in Figure 1 regarding the location of the network $\psi$. We will update the figure with the correct notation to match the expression in the manuscript.
> #### 4. Positioning with Recent Literature
>
> We appreciate your recommendations on relevant recent works, including *LaBO*, *BotCL*, and *ResCBM*.
>
> - The suggested works leverage Vision-Language Models (VLMs), such as CLIP, by passing images to the visual encoder and directly using textual concepts (in text format) as input to the language encoder. These models ensure alignment between the visual and textual concepts to form a concept bottleneck layer, which is then used for the final classification task.
> - **MIMOSA’s distinct approach**:
>   - MIMOSA’s text encoder operates on input texts that contain information not only related to the concepts but also to the task (see Figure 1, for instance; the task can be inferred by looking at both the image and textual inputs).
>   - The concepts here are predicted through a shared representation of both input modalities in a supervised manner through the concept embedding layer.
>
> This distinction allows MIMOSA to address use cases, such as medical imaging with broader text annotations or audio-video classification, where other models are limited.
>
> We will clarify this distinction and highlight the novelty of integrating multiple modalities in a concept-based framework in the revised manuscript.
> #### 5. Limited Dataset Scope
> We acknowledge the importance of diverse dataset evaluation and are working to expand the experiments.
> - We would like to kindly mention that the CUB dataset has already been included in our experiments.
> - Also, the CelebA, CheXpert, and AWA2 datasets are not suitable for the proposed methodology as they only provide input images and lack additional modalities such as text. We found a variant of CelebA that provides textual captions and might be suitable for expanding the experimental campaign.

---

> ### Author Response · Authors · 2024-11-20
> **Response to the reviewer pUyV (2/2)**
>
> #### 6. Text data source clarification:
> To improve transparency, we will provide detailed descriptions of the text data source and preprocessing steps in the Appendix:
>
> 1. **MNIST+**:
>     - Each image pair is assigned a caption randomly generated by varying a range of templates.
>     - These templates are created through multiple variations of digit descriptions and concatenation of descriptive sentences.
>     - Specifically, we defined 10 unique descriptions for each digit (example for 5: “the number of fingers on one hand”) and 10 concatenations of descriptive sentences (example of a sentence: “the first digit is [description1] and the second [description2]”), potentially creating a different sentence for each image (a couple of digits) available in the dataset.
>
>
> 2. **cdSprites+**: Textual descriptions are directly included in the original dataset alongside predefined attributes serving as concepts.
>
> 3. **CUB**: We started from the extended dataset version incorporating 10 descriptive captions paired with the bird images [^1] .
> - During the dataset preparation, each caption's quality was revised (check for unknown words or duplicates) to ensure that only accurate captions accompanied each image.
> - Then, the alignment with the original dataset[^2] containing the concepts was performed to ensure that each image was paired with its corresponding caption and associated concepts.
> - This process resulted in an enhanced version of the CUB dataset, with carefully aligned images, corresponding concepts, and captions.
>
> We are planning to release these datasets.
>
> #### 7. Expanded model comparisons
> MIMOSA’s primary objective is to compare it against state-of-the-art unimodal concept-based interpretable by-design models (*CBM*, *CEM*).
> - Comparison with CBM and CEM:
>   - CBM and CEM models were evaluated in their standard visual-only setting and with text-only setting following two recent contributions (Tan et al., 2024; De Santis et al., 2024), and we evaluated their performance as baselines.
> - Comparison with E2E:
>   - We compared MIMOSA against an end-to-end (E2E) model that does not have a concept layer and so does not rely on concept predictions to do the final task. We did this comparison to highlight the value of interpretability.
> - Comparison with *SHARCS*:
>   - This comparison is proposed since SHARCS employs multimodal inputs and relies on concept-based interpretability. However, this has separate concept representations for modalities, unlike MIMOSA’s shared concept space.
>
> Methods like *LaBO* and *Post-hoc CBM* were not included, as they use concepts themselves as textual input rather than broader textual inputs such as captions of the images.
> Our focus with MIMOSA is on leveraging textual descriptions and visual data to generate shared, interpretable concept representations rather than directly inputting predefined concepts. This distinction aligns with our goal of developing a unified framework for multimodal interpretability.
>
> We will highlight this choice in the revised manuscript.
> #### 8. Negative Intervention Experiment
> We appreciate this insightful suggestion. To present a more comprehensive evaluation of MIMOSA, we are incorporating negative intervention experiments. This addition will provide further insights into MIMOSA’s robustness and behavior under varied intervention scenarios.
>
>
> ---
> Thank you again for your detailed feedback. We believe that these updates will strengthen our paper and enhance its clarity, and we look forward to addressing any further questions you may have.
>
>
> ---
>  [^1]: https://www.kaggle.com/datasets/coolerextreme/cub-200-2011
> [^2]: https://www.vision.caltech.edu/datasets/cub_200_2011/

---

> > ### Comment · Reviewer_pUyV · 2024-11-20
> > **Thanks for your responce**
> >
> > Thank you for your thoughtful response. After careful consideration, I have decided to maintain the scores I initially provided.

---

> > > ### Author Response · Authors · 2024-11-21
> > > **Reply to reviewer pUyV**
> > >
> > > Dear Reviewer pUyV,
> > >
> > > If there are specific aspects or critical points that you find problematic, we would be more than happy to provide further clarification or consider incorporating necessary improvements, either in this submission or in the future.
> > >
> > >
> > > We genuinely believe that the contributions of our work address significant challenges in the field, and your suggestions for improvement would help us in refining it.
> > >
> > > ---
> > > Thank you for your time and effort.

---

### Official Review · Reviewer_arW7 · 2024-10-30

**Soundness:** 1
**Presentation:** 1
**Contribution:** 1
**Rating:** 1
**Confidence:** 4

**Summary:**

In this paper, the authors presented MIMOSA, a multimodal model that encodes each modality with unimodal encoders, fusion with addition, and has an additional reconstruction loss. MIMOSA was evaluated against baselines on 3 tasks, where MIMOSA achieved top performance in none of the tasks. The authors also evaluated MIMOSA's conceptual accuracy and concept mismatch rate, and qualitatively demonstrated that the decoders can reconstruct the input.

**Strengths:**

None.

**Weaknesses:**

This paper has no significant contribution. The proposed method is basically a simple multimodal model with unimodal encoders, fusion by summation, and reconstruction loss with decoders. Multimodal reconstruction loss already exist in many existing papers (e.g. MVAE, which is actually mentioned in the paper). 2 out of 3 datasets used in the experiments are really simple where multiple methods achieve near-perfect performance; and the proposed method did not achieve top performance in any of the datasets. The proposed method also does not improve model explainability at all. Being able to reconstruct inputs from multimodal representation (section 4.4) is not explainability; it just means the reconstruction loss worked. The so-called "concept" generated by MIMOSA is just a vector representation of the multimodal input, with no objective making it any more interpretable than usual autoencoder-generated representation vectors. Therefore, there is no clear takeaway message from the paper.

This paper also has a lot of important details missing, which makes interpretation of results and reproducing the method impossible. The missing information includes (but not limited to):

1. Full training objective; which parts of MIMOSA is trainable and which ones are frozen; how are decoders trained

2. How Stable Diffusion can be used (and trained) as an image reconstruction decoder

3. Evaluation metrics: there is no concrete definition of Concept Accuracy and Concept mismatches, which makes the majority of results in sections 4.1 and 4.2 uninterpretable.

4. How the experiments were repeated (to get the Standard Deviations in the tables), and what learning rate/optimizer was used

5. Baseline details: there is no description of what exactly E2E is at all, even though it got highest performance on CUB by a large margin. There is also no detailed hyperparameters about any of the baselines.

The paper also has the following additional presentation issues:

1. Inconsistent notations: the decoders are notes as $\Psi_i$ in text and $\bar{h_i}$ in Fig 1. Also, in line 183, the exact same variable is written differently ($\Psi(c)$ and $\Psi_i(c)$)

2. Dataset name inconsistency: the second dataset is called differently ("cdSprites" and "cdSprites+") multiple times across many locations in the paper ("cdSprites" in Fig 3 caption and Fig 6 caption and line 242, "cdSprites+" for other 13 occurrences)

3. It is unclear why the text is aligned in different directions among the subfigures in Fig 4

4. In Fig 2, it is really strange to draw both models and vectors with identical circles

**Questions:**

1. What is the overall training objective of MIMOSA? The only objective mentioned in the paper is $L_{dec}$, which can't possibly be the only objective used to train the model. You at least need the supervised task objective. How is $L_{dec}$ combined with the supervised objective?

2. Which parts of MIMOSA is being trained and which parts are frozen? Do you jointly train all your unimodal encoders with the rest of the model? How do you jointly train a Stable Diffusion model with the other parts of your model?

3. How exactly are the standard deviations obtained for your experiments? Please provide details on whether you repeated your experiments with different seeds, and for how many times. The standard deviations for a lot of entries in the table looks unreasonably low.

4. Please also provide the other hyperparameters for your experiments, such as learning rates and baseline implementations.

5. In Eq(1), how does a summation of unimodal representations (since $\Phi$ is simple summation operator) yield 2 outputs? And what exactly do you do with $\hat{c}$?

---

> ### Author Response · Authors · 2024-11-20
> **Rensponse to reviewer arW7 (1/2)**
>
> Dear Reviewer arW7,
>
> Thank you for your valuable feedback.
>
> To address the concerns raised, we would like to start by emphasizing **our contributions**:
>
>
> 1. **Multimodal interpretable by-design concept-based model**
>    *MIMOSA* is an interpretable by-design concept-based model, as it allows for the interpretation of its predictions through concept prediction. Specifically, users can identify which concepts contributed to the final task prediction by inspecting the concept predictions. This approach aligns with the principles of interpretable by-design concept-based models such as Concept Bottleneck Models (CBMs) introduced by Koh et al. (2020) and Concept Embedding Models (CEMs) introduced by Espinosa Zarlenga et al. (2022). These models explicitly use intermediate concepts to influence task predictions, providing built-in interpretability without relying on post-hoc explanations.
>
>    Unlike traditional concept-based models, which typically focus on a single input modality, *MIMOSA supports multiple modalities*, incorporating both textual and visual inputs, bridging an important gap in current concept-based explainability approaches.
>
> 2. **Novel interpretability through concept prototype visualization**
>    We also provide another form of interpretability via concept prototypes. Current concept-based models are unable to visualize the concepts learned. Instead, *MIMOSA* allows for visualization of the concepts the model relies on to make the final prediction by projecting the concepts.
>
> 3. **Shared concept embedding representation**
>    A previous attempt to create a multimodal CBM was proposed in SHARCS by Dominici et al. (2023). However, SHARCS is limited in its employment since it uses a separate concept representation. On the contrary, *MIMOSA* employs a shared embedding space for concepts derived from both modalities. This allows a unique concept representation that is valid across modalities, more interveneable, and allows a complete concept prototype visualization in both modalities.
>
> 4. **SOTA performance among interpretable methods with single concept representation**
>    While *MIMOSA*’s primary focus is interpretability, it achieves state-of-the-art accuracy among CBMs with single concept representation, with only a minimal gap compared to the SHARCS and end-to-end (E2E) models (that does not rely on concepts). Additional analyses, such as the impact of interventions on task predictions and qualitative visualizations of concept prototypes, further demonstrate its capability.
>
> ---
> ### Responses to Specific Concerns
>
> #### 1. Presentation Problems
> Thank you for pointing out the presentation errors. They will all be addressed in the revised version.
>
> #### 2. Detailed Model Architecture
> We appreciate your suggestion to provide further details about the model structure. Below is a breakdown of the *MIMOSA* architecture:
>
> - **Encoders:**
>   - *Image Encoder*: Utilizes a pre-trained ResNet50 backbone.
>   - *Text Encoder*: Initialized with a pre-trained language model (BERT base for MNIST+ and cdSprites+ datasets, BERT large for CUB).
>
> - **Concept Embedding Layer:**
>   - Based on the implementation from Espinosa Zarlenga et al. (2022) (CEMs).
>   - Trained from scratch using binary cross-entropy loss for concept predictions.
>
> - **Decoders:**
>   - For MNIST+/cdSprites+:
>     - Image decoders are convolutional neural networks trained from scratch. Losses tested include Mean Squared Error (MSE), L1 distance, and Structural Similarity Index (SSIM).
>     - Text decoders compute cross-entropy loss over predicted captions.
>   - For CUB:
>     - Image decoder employs a pre-trained Stable Diffusion v1-4 model. Unlike conventional usage, where the model is prompted with text for inference, we fine-tune the decoder using the concept embeddings generated by MIMOSA. These embeddings act as a representation of concepts in a latent space. This fine-tuning allows Stable Diffusion to visualize the concepts directly, even without textual prompts. The fine-tuning process uses SSIM as the primary loss to ensure alignment between visualized concepts and the input data.
>
> These details will be included in the revised paper.
>
> #### 3. Training Objectives
> *MIMOSA* uses the following objectives:
>
> - **Concept Prediction Loss:** Binary Cross Entropy Loss for supervised concept predictions.
> - **Task Loss:** Cross-Entropy loss for the final classification task.
> - **Reconstruction Loss:** Modality-specific losses (e.g., MSE and SSIM for images, Cross-Entropy for text) ensure the alignment of shared concept embeddings with input modalities.
>
> The overall training objective combines these losses:
>
> $
> \text{L} = \lambda_\text{task} L_\text{task} + \lambda_\text{concept} L_\text{concept} + \lambda_\text{dec} L_\text{dec}
> $

---

> ### Author Response · Authors · 2024-11-20
> **Rensponse to reviewer arW7 (2/2)**
>
> #### 4. Hyperparameter Specification
>
> | HP              | MNIST+          | cdSprites+      | CUB             |
> |------------------|-----------------|-----------------|-----------------|
> | Learning Rate $lr$ | $1\text{e-3}$, $1\text{e-4}$ (with decoders) | $1\text{e-3}$, $1\text{e-4}$ (with decoders) | $1\text{e-3}$, $3\text{e-4}$ (with decoders) |
> | $\lambda_\text{task}$ | 0.1             | 0.1             | 0.1             |
> | $\lambda_\text{concept}$ | 1               | 1               | 1               |
> | $\lambda_\text{dec}$  | 1               | 10              | 10               |
> | Epochs          | 50              | 50              | 50              |
>
> #### 5. Training and Freezing Components
>
> - **Trainable Components:** All components, including encoders, decoders, concept embedding layer, and task predictor, are trainable. For the CUB dataset, Stable Diffusion is fine-tuned on concept embeddings.
> - **Frozen Components:** Only for MNIST+ and cdSprites+, we use the text encoder frozen as on these datasets, the extracted representation of the last attention layer is already sufficiently rich.
>
> #### 6. Reproducibility of Experiments
> Experiments were conducted three times with different seeds. Results represent averages with standard deviations to reflect variability. Long training ensured the convergence of the decoder models, producing consistent results.
>
> #### 7. Equation 1 Problems
> Thank you for highlighting the typos in Equation 1. These will be corrected as follows:
>
> - $\oplus$ represents the pooling operator (sum in our case).
> - $\phi$ denotes the concept embedding layer, which generates both concept predictions $\hat{c}$  and concept embeddings.
>
> ---
>
>
> Thank you again for your detailed feedback. We believe that these updates will strengthen our paper and enhance its clarity, and we look forward to addressing any further questions you may have.

---

> > ### Comment · Reviewer_arW7 · 2024-11-20
> > **Further Questions**
> >
> > Thank you for your reply! Given your response, I still have several questions:
> >
> > 1. How do you train a Stable Diffusion model with SSIM? Stable Diffusion uses latent diffusion, which means all diffusion happens in the latent space, and the training objective is usually L2 loss between the target epsilon and predicted epsilon. Computing SSIM on the latent noise doesn't make much sense. In addition, you mentioned in point 5 that you jointly fine-tune this decoder with the rest of MIMOSA, and it is unclear to me how it is possible to jointly fine-tune a diffusion-based decoder (where the image generation process can take many iterations of reverse diffusion) and the rest of the classification model.
> >
> > 2. Can you provide more details about the Concept Prediction loss (since there wasn't any in the original paper)? How exactly is it computed with $\hat{c}$ or **c**, and what labels are used to supervise them?
> >
> > 3. What exactly do you mean by "projecting the concepts"? The term "projecting" wasn't in the main paper. It is also unclear how that helps with interpretability. Since you trained the model with reconstruction loss, this just shows that you can reconstruct input from intermediate representation. Can you give a more detailed explanation on why MIMOSA provides better interpretability? Especially compared to a conventional multimodal model with late fusion and reconstruction loss added to it.
> >
> > 4. Can you provide more information on the Baselines? Especially E2E model, since there is no description about its architecture or training objective in the main paper.

---

> > > ### Author Response · Authors · 2024-11-21
> > > **Response to Further Questions of reviewer arW7 (1/2)**
> > >
> > > Dear Reviewer arW7,
> > >
> > > Thank you for your detailed follow-up questions and for taking the time to engage with our work so thoroughly. We appreciate your feedback and would like to address your concerns point by point.
> > >
> > > ---
> > > #### **1. Stable Diffusion and training**
> > > - Stable Diffusion loss: To clarify, in MIMOSA, we do the fine-tuning process to include an SSIM-based loss applied to the decoded image outputs rather than the latent noise.
> > > This adjustment ensures the SSIM loss is computed where it is semantically meaningful - on the reconstructed images- and it is computed between the input image and the reconstructed one.
> > > - Training Stable Diffusion model: We understand your concern, as fine-tuning a diffusion-based decoder jointly with a classification model is indeed non-trivial. Here is a more detailed explanation of our approach.
> > > In MIMOSA we employed a multi-stage fine-tuning strategy:
> > >    - Initially, we froze the pre-trained stable diffusion decoder during the first stage of training (also the text decoder). During this phase, only the classification components of MIMOSA and the concept embedding layer were trained, ensuring stability while allowing the model to establish meaningful intermediate concept representations based on pre-trained latent space mappings.
> > >    - Once the classification components reached a reasonable performance level, we fine-tuned the diffusion decoder with SSIM loss, as explained above, while keeping the classification model fixed.
> > >
> > > This strategy allowed us to decouple the optimization dynamics of the diffusion decoder from the rest of MIMOSA, avoiding instability while progressively aligning both components. We will include these clarifications in the revised paper.
> > >
> > > #### **2. Concept Prediction Loss**
> > > Concept Prediction Loss is designed to enforce alignment between the model’s intermediate representations and human-understandable concepts.
> > > Below, we provide a more detailed definition and computation method, along with a toy example:
> > >
> > > - Definition: The Concept Prediction Loss measures the discrepancy between the predicted intermediate concept representations and their corresponding ground-truth values, which can be either explicitly provided in the dataset or derived from auxiliary annotations. In MIMOSA, we already have ground-truth annotations of concepts.
> > > - Computation of $\hat{c}$ and **c**: In MIMOSA, the concept predictions $\hat{c}$ and concept embeddings **c** are calculated with the Concept Embedding Layer as proposed by Espinosa Zarlenga et al. (2022)[^1]
> > > - Loss: In MIMOSA, we use a binary cross-entropy (BCE) loss to quantify the discrepancy between the model's predicted concept values and the corresponding ground-truth annotations, ensuring accurate concept prediction.
> > >
> > > [^1]: https://proceedings.neurips.cc/paper_files/paper/2022/hash/867c06823281e506e8059f5c13a57f75-Abstract-Conference.html
> > >
> > > We provide a toy example:
> > > Suppose the possible concepts in the dataset are **size** (e.g., *big*, *small*), **shape** (e.g., *square*, *circle*), and **color** (e.g., *blue*, *pink*).
> > >
> > > We have an input image depicting a **big blue square** with the caption *"A big, blue square"*.
> > > The concepts for this multimodal input are:
> > > - **Size:** *big*
> > > - **Shape:** *square*
> > > - **Color:** *blue*
> > >
> > > The ground-truth concept vector $ \bar{c}$ will be a binary vector representation where each concept that is present in the input is marked with a value of 1, and those that are absent are marked with 0. For this example, the vector would look like:
> > >
> > > $\bar{c} = [\text{big}=1, \text{small}=0, \text{square}=1, \text{circle}=0, \text{blue}=1, \text{pink}=0]$
> > >
> > > The output of the **concept embedding layer** comprises two key components:
> > > 1. **Concept embeddings $c$:** These are vector representations capturing the semantic meaning of each concept in the latent space.
> > > 2. **Concept predictions $\hat{c}$:** These are the model’s predictions about the presence or absence of each concept for the given input. In this case, the model should predict something close to:
> > >
> > > $\hat{c} = [\text{big}=0.95, \text{small}=0.05, \text{square}=0.90, \text{circle}=0.10, \text{blue}=0.98, \text{pink}=0.02]$
> > >
> > > The **Concept Prediction Loss** is then computed as the **Binary Cross-Entropy (BCE)** between the ground-truth vector $\bar{c}$ and the predicted concept values $\hat{c}$ for all concepts:
> > >
> > > $\text{Loss} = \frac{1}{N} \sum_{i=1}^N \Big( \bar{c}_i \log(\hat{c}_i) + (1 - \bar{c}_i) \log(1 - \hat{c}_i) \Big)$
> > > where $N$ is the total number of concepts in the dataset.
> > >
> > > #### **3. Use of term projecting**
> > > We recognize that the term *projecting* may be misleading. By projection, we mean the **visualization** (as image or text) of concepts. Visualizing what the model has learned as concepts is a new form of interpretability that allows us to explicitly *see* intermediate representations of the model through meaningful concepts.

---

> > > ### Author Response · Authors · 2024-11-21
> > > **Response to Further Questions of reviewer arW7 (2/2)**
> > >
> > > #### **4. Improved Interpretability in MIMOSA**
> > > - Limitations of Conventional Multimodal Models with Reconstruction Loss: Conventional multimodal models using late fusion and reconstruction loss focus on reconstructing inputs from latent representations. While reconstruction loss ensures these representations retain relevant information, it does not make them interpretable because the latent features are not aligned with human-understandable concepts.
> > >  This lack of alignment makes it challenging for humans to understand the inner workings of such models.
> > > - How Mimosa Enhances Interpretability:
> > >    - Concept Mapping: MIMOSA explicitly maps intermediate representations to **human-understandable concepts** through its concept embedding layer. These concepts form the sole input to the final classification task, ensuring that predictions are based on interpretable units.
> > >  For example, in a bird species classification task, MIMOSA can provide explanations like *“This bird has been classified as a specific breed because it is small, has a curved beak, and is blue.”*
> > > Additionally, MIMOSA visualizes concepts by reconstructing them through its decoder. This provides users with a tangible view of what the model "understands" about the data, unlike conventional models, which only reconstruct inputs without offering insights into the model’s reasoning.
> > >    - Early Fusion for Consistent Concepts: MIMOSA integrates modalities early in the pipeline, creating a unified representation before mapping to the concept space. This ensures that derived concepts are consistent and free from modality-specific conflicts.
> > >  Our experiments demonstrated the challenges of late fusion (e.g., as used in SHARCS by Dominici et al., 2023). In late-fusion concept-based models, modalities independently predict features, leading to **concept mismatches** when predictions conflict. These mismatches hinder interpretability and task reliability.
> > >  Early fusion in MIMOSA eliminates such conflicts, ensuring a cohesive reasoning process grounded in consistent and interpretable concepts.
> > >
> > > #### **5. E2E Model**
> > > Thank you for pointing out the lack of information about the E2E baseline in the main paper. Below, we provide a comprehensive description of its architecture and training objective:
> > > -  **Architecture**:
> > >    - The E2E model utilizes the same encoders described for MIMOSA:
> > >        - Image encoder: ResNet50
> > >        - Text encoder: BERT encoder (BERT base for MNIST+ and cdSprites+ datasets, BERT large for CUB)
> > >     - The outputs of these encoders are **summed** to create a unified representation that combines the image and text modalities.
> > >     - This combined representation is then passed to the **task predictor**, a feedforward neural network comprising:
> > >        - Two linear layers separated by a LeakyReLU activation function and dropout for regularization to prevent overfitting.
> > >     - The task predictor maps the summed multimodal representation directly to the task predictions.
> > > - **Training Objective**: The E2E model is trained solely using the task loss $L\_{task}$, which directly optimizes for the downstream prediction task (e.g., classification). Unlike MIMOSA, the E2E model does not utilize intermediate concept supervision or reconstruction objectives, making it a simpler yet less interpretable baseline.
> > >
> > > This architecture serves as a straightforward end-to-end learning framework, but it lacks the interpretable intermediate steps provided by MIMOSA. We will include this detailed description in the revised version of the paper.
> > >
> > >
> > > ---
> > >
> > > Thank you again for your follow-up questions. We hope this clarification addresses your concerns. Please let us know if you have additional questions or would like further details.

---

> > > > ### Comment · Reviewer_arW7 · 2024-11-25
> > > >
> > > > Thank you for your response! You have provided many important details in the rebuttals that are absolutely crucial for understanding and reproducing the model architecture and experiments, but all of these information are not in the main paper. Therefore, I believe the current state of the paper is not fit for publication. In addition, I am still skeptical of the significance of the "interpretability" gained from supervised concept training, as it seems to require a lot of additional annotations and does not generalize to most real datasets where concept annotations are unavailable. It seems closer to "jointly training main task and relevant sub-tasks" than interpretability. Thus, my score remains unchanged.

---

### Official Review · Reviewer_YUpA · 2024-11-01

**Soundness:** 2
**Presentation:** 1
**Contribution:** 1
**Rating:** 1
**Confidence:** 5

**Summary:**

The paper introduces MIMOSA (MultIMOdal concept-based repreSentAtions), a novel unified multimodal model that integrates multimodal representation learning with concept-based interpretability techniques. MIMOSA is primarily focused on text and image modalities, aiming to enhance the interpretability of deep learning models by extracting concept prototypes from a shared embedding space.

**Strengths:**

Based on the traditional interpretable conceptual model, this paper proposes a multi-dimensional interpretable method, which uses image decoding and text decoding strategies to provide multi-dimensional interpretable strategies for the source data, and provides a new interpretable perspective.

**Weaknesses:**

1) The experimental data are not convincing. In Table 1, the method proposed in this paper does not maintain a high level and cannot reflect the superiority of multidimensional interpretability. Secondly, in terms of multidimensional interpretability, this paper does not make a clear contrast with the existing methods on the interpretation Angle, and does not clearly reflect the meaning of multidimensional.

2) This paper proposes multi-dimensional interpretable methods, the purpose is not clear, is it to improve interpretability, improve robustness, or improve task performance? Purpose 3 should be excluded first, because the proposed method does not outperform the existing models in the data set of this paper. Secondly, the improved interpretability lacks demonstration and fails to well reflect the interpretability perspective. Ultimately, for robustness, there is a lack of a specific description of the intervention. Is the intervention performed on the input image, the input text, or some other input source?

3) Figure 3 should have errors, task accuracy should not rise as the intervention is enhanced, here should be drawing errors.
The training details of this paper need to be further clarified. How is it trained for the added text description? How is it applied against the existing concept data, and how is it aligned against the generated image decoder? Further clarification is needed.

4) For the experiment of intervention, it is necessary to supplement the intervention of image and the intervention of input text, so as to achieve a more comprehensive response robustness. However, this paper lacks relevant descriptions.

5) No ablation experiments were performed in this paper, so the specific role of each module cannot be clarified.

6) For the CUB200 dataset, how to carry out the text supplementary description? Further clarification is needed.

7)How to evaluate the concept accuracy in Table2? The concept structure of interpretable description generated by this method is different from that of data annotation. How to evaluate it?

8) For Table1, why design a txt only benchmark? What does txt alone say? Moreover, for CBM and CEM models, without the input of image, how can the accuracy rate be obtained?

**Questions:**

see weaknesses.

---

> ### Author Response · Authors · 2024-11-20
> **Response to reviewer YUpA (1/2)**
>
> Dear Reviewer YUpA,
>
> Thank you for your valuable feedback.
>
> Below, we provide responses to the specific concerns raised:
>
> ---
> #### 1. Multimodal Interpretability and Purpose of MIMOSA
> The primary goal of MIMOSA is to create a multimodal model based on explainable concepts, aiming to strike a balance between interpretability and accuracy. While achieving state-of-the-art predictive performance is valuable, it is not the central focus of this work. MIMOSA offers a novel approach by *extending concept-based models to the multimodal domain*, seamlessly integrating both textual and visual explanations into a unified framework. This design prioritizes understanding and transparency, recognizing that interpretability might sometimes influence accuracy.
>
> MIMOSA is inherently interpretable by design, allowing users to trace task predictions back to the contributing concepts. This approach builds on the principles of Concept Bottleneck Models (CBMs) and Concept Embedding Models (CEMs), where intermediate concepts are explicitly used to influence predictions, ensuring interpretability without reliance on post-hoc explanations. The core purpose of MIMOSA is to enhance interpretability in multimodal settings through its concept-based model while maintaining competitive accuracy.
>
> We will clarify this in the revised manuscript.
>
> #### 2. Concept Interventions:
> Concept Interventions are applied at the concept level, following the approach proposed by Koh et al. (2020), where no modifications are made to the input features, as altering them may not be interpretable for domain experts.
>
> These interventions are reported in terms of task accuracy improvements, as they directly influence predictions. The accuracy increases with intervention because the model benefits from progressively more accurate concept corrections, which improve prediction performance. This behavior simulates an expert interacting with the model—acting as an oracle—by correcting predictions and testing if the model can recover the correct task prediction.
>
> Additionally, MIMOSA introduces a unique perspective by visualizing the effects of concept interventions at the decoder output, thanks to its concept visualization with decoders. We also plan to supplement this work with experiments involving negative interventions on both image and text inputs, as suggested by Reviewer pUyV.
>
> #### 3. Ablation Experiments
> Ablation studies are indeed very important in understanding the contribution of each part of a model. While we agree that further studies could have been conducted, we already presented two ablation studies in the paper. We tested different multimodal concept representations for our model depending on the concept layer adopted (CBM-Linear, CBM-MLP, and CEM).
>
> We also compared MIMOSA with different unimodal versions of CBM-Linear, CBM-MLP, and MIMOSA, both for image and textual data.
>
> This experimental analysis aimed to assess whether the improved quality of MIMOSA stems from leveraging one specific modality or from integrating text as an additional mode.
>
> The results demonstrated that the inclusion of textual data, while alone it is not sufficient for competitive performance, contributes to the model’s overall performance. In the revised manuscript, we will expand on these findings and include further ablation studies on other components, such as the pooling mechanism, to provide a more comprehensive analysis of the factors influencing MIMOSA’s performance and interpretability.
>
> #### 4. Clarification on textual descriptions in CUB
> To implement the CUB dataset with captions and concepts we:
> -Started from the extended dataset version incorporating 10 descriptive captions paired with the bird images [^1] .
> - During the dataset preparation, each caption's quality was revised (check for unknown words or duplicates) to ensure that only accurate captions accompanied each image.
> - Then, the alignment with the original dataset[^2] containing the concepts was performed to ensure that each image was paired with its corresponding caption and associated concepts.
> - This process resulted in an enhanced version of the CUB dataset, with carefully aligned images, corresponding concepts, and captions.
>
>
> ---
>  [^1]: https://www.kaggle.com/datasets/coolerextreme/cub-200-2011
> [^2]: https://www.vision.caltech.edu/datasets/cub_200_2011/

---

> > ### Comment · Reviewer_YUpA · 2024-11-22
> >
> > Thanks for your feeble reply,
> >
> > 1. You mentioned that revisions would be made in the revised manuscript, but I have not seen any modifications in the current draft.
> >
> > 2. The author(s) still fail to address the lack of demonstrable interpretability of the additional imaging modality, which does not adequately highlight the aspect of interpretability.
> >
> > 3. Corresponding to point 2, if there is no performance improvement, the work on interpretability should strive towards greater automation. However, this work introduces additional manual annotations, significantly reducing its innovation.
> >
> > 4. I do not understand the significance of explainability when images are fed into a processor and then reconstructed through a decoder to produce a "similar image." If this is the case, why not simply freeze the original image feature extraction layer and output the original feature maps, which would be clearer and more interpretable? This point is highly contradictory, and I find it puzzling why such an image-based approach to explainability is employed, especially since it seems to be the main innovation of the paper. Even with the integration of image explainability, there is no enhancement in the performance of the original task, making the image layer perplexing.
> >
> > 5. There is a lack of detailed description regarding the interventions, how they were conducted, what the input and output examples are, and exactly what interventions this work has performed. Do not directly cite examples from previous works, because you have introduced a multimodal approach, which is a completely new framework. If the intervention aspects are identical to those in previous works, is your framework innovative? This is particularly evident in the appendix, where the content is too sparse and does not resemble a scholarly paper but rather an unfinished arXiv report.

---

> ### Author Response · Authors · 2024-11-20
> **Response to reviewer YUpA (2/2)**
>
> #### 5. Concept accuracy in Table 2
> Concept Accuracy refers to the proportion of correctly predicted concepts in a model's output compared to the ground truth. It measures how well the model can identify or predict the high-level, interpretable features (concepts) associated with the input data.
>
> The evaluation of concept accuracy in Table 2 measures the model’s ability to predict predefined concepts based on concept ground-truth annotations correctly. Notice that the concept prediction does not entail the decoders, whose purpose is only to provide a visual representation of the concepts and whose annotations are represented by the same input modalities.
>
> We will clarify this in the revised paper.
>
> #### 6. Text-only benchmark in Table 1
> - The text-only benchmark was included to evaluate the independent contribution of the textual modality. This analysis highlights the role of textual data in concept-based multimodal learning.
> We will make this point clearer in the revised manuscript.
>
> - CBM and CEM were natively designed for image data. However, we followed the architectural modifications and text processing methods described in two recent contributions (Tan et al., 2024; De Santis et al., 2024), and we evaluated their performance as baselines also with textual inputs. We did this as an ablation study to verify which of the two modalities alone performed better.
>
> We will further clarification on these experimental setups to avoid confusion.
>
> ---
>
> Thank you again for your detailed feedback. We believe that these updates will strengthen our paper and enhance its clarity, and we look forward to addressing any further questions you may have.

---

### Official Review · Reviewer_nZCv · 2024-11-03

**Soundness:** 2
**Presentation:** 2
**Contribution:** 2
**Rating:** 3
**Confidence:** 4

**Summary:**

This paper applies concept bottleneck models to multimodal data, proposing the MIMOSA (MultIMOdal concept-based representations) approach. The method is composed of a concept layer and multimodal decoders. The concept layer aggregates the modality-dependent encoders' representations and generates concept prediction scores and embeddings. Decoders reconstruct the inputs and visualize concepts explicitly. The experimental results show enhanced accuracy and interpretability compared to other unimodal methods.

**Strengths:**

- The paper is well-written and provides a clear exposition of the background, related work and proposed method.
- Concept-based interpretability is a promising research topic. Previous methods mainly focused on visual data. This paper extends the concept-based models to multimodal data, which is interesting and advantageous.
- The method in this paper is very intuitive and easy to understand.

**Weaknesses:**

- The proposed method lacks innovation. There is little difference between CEM and MIMOSA, and the decoding-based visualization strategy is also commonly used.
- In this paper, the author only utilizes the limited-scale datasets, not validating the efficiency of large-scale multimodal datasets. I think at least one large dataset should be conducted experiments to demonstrate the scale-up capacity of the proposed method.
- On MNIST+ and cdSprites+, the proposed method's improvements are very few.
- This paper doesn't demonstrate the ablation studies. It needs more discussions on what module contributes to the improvement of accuracy.

**Questions:**

- I am not sure about the definition of the concept in this paper. The concepts for MNIST+ are more likely a group of descriptions.
- I just see the decoding visualization results at the image level. Decoding at the image level rather than at the single concept level lacks persuasion. I wonder what the results would be if we focused on one single concept.
- CBM and CEM are proposed for only visual data input, so I am unsure whether the comparisons are fair enough.
- This paper seems to apply CEM to multimodal data using a simple sum operator, lacking a delicate design for the specific setting. Moreover, the ablation studies of other pooling operators need to be done.

---

> ### Author Response · Authors · 2024-11-20
> **Response to reviewer nZCv**
>
> Dear Reviewer nZCv,
>
> Thank you for your valuable feedback.
>
> Below, we provide responses to the specific concerns raised:
>
> ---
> #### 1. Innovations and Distinctions from CEM
> MIMOSA introduces several innovations specific to the multimodal setting.
> - Unlike CEM, MIMOSA aggregates information from multiple modality-dependent encoders to form a shared concept embedding layer, integrating cross-modal information.
> - Additionally, MIMOSA enables the visualization of concept prototypes. Unlike standard decoders, MIMOSA’s decoders initiate the reconstruction process directly from the shared concept representation space rather than from the modality-specific encodings of the input data.
> - Furthermore, to the best of our knowledge, MIMOSA is the first supervised concept-based model to employ decoders. Decoders are typically used in unsupervised concept-based models to learn disentangled concept representations. Instead, we propose using them on top of a supervised concept representation to visualize it.
>
> We will clarify these distinctions more explicitly in the revised manuscript.
>
> #### 2. Performance on MNIST+ and cdSprites+
> The improvements of MIMOSA on MNIST+ and cdSprites+ are indeed modest.
> However, the goal of MIMOSA is not solely to enhance performance but to increase interpretability through concept-based explanations across multiple modalities.
> For example, on cdSprites+, MIMOSA can identify and explain factors like "object shape", "color," and "position" that influence predictions.
>
> #### 3. Ablation studies
> Ablation studies are indeed very important in understanding the contribution of each part of a model. While we agree that further studies could have been conducted, we already presented two ablation studies in the paper. We tested different multimodal concept representations for our model depending on the concept layer adopted (CBM-Linear, CBM-MLP, and CEM).
>
> We also compared MIMOSA with different unimodal versions of CBM-Linear, CBM-MLP, and MIMOSA, both for image and textual data.
>
> This experimental analysis aimed to assess whether the improved quality of MIMOSA stems from leveraging one specific modality or from integrating text as an additional mode.
>
> The results demonstrated that the inclusion of textual data, while alone it is not sufficient for competitive performance, contributes to the model’s overall performance. In the revised manuscript, we will expand on these findings and include further ablation studies on other components, such as the pooling mechanism, to provide a more comprehensive analysis of the factors influencing MIMOSA’s performance and interoperability.
>
> #### 4. Definition of Concepts in MNIST+
> In MIMOSA, the input consists of images, their textual descriptions, and associated concepts.
> In the case of MNIST+, the digits themselves serve as the concepts, while the descriptions are processed by the textual encoder. As an example, we can look at Figure 4 - Original. In this example, the concepts are 4 and 3, while the textual description is the one reported below.
>
> We will clarify this setup in the manuscript to make the distinction between concepts and descriptions more explicit.
>
> #### 5. Comparisons with CBM and CEM
> In the revised manuscript, we will clarify that the comparisons between MIMOSA, CBM, and CEM are intended to establish a baseline for concept-based interpretability rather than to conduct a direct, one-to-one performance evaluation.
>
> While CBM and CEM were originally designed for image data, we adapted them to handle textual inputs by following the architectural modifications and text processing methods outlined in recent works (Tan et al., 2024; De Santis et al., 2024). We evaluated these modified versions as baselines to assess the performance of each model when applied to textual data. This ablation study was conducted to investigate the performance of the models with individual modalities (image vs. text) and to determine which modality, when used alone, led to better performance.
>
> #### 6. Pooling Operator choice and Ablation studies:
> Thank you for the suggestion. In our ablation studies, we are adding additional pooling strategies (e.g., max pooling, concatenation) to analyze their effect on performance and will include this analysis in the revised draft.
>
>
> ---
> Thank you again for your detailed feedback. We believe that these updates will strengthen our paper and enhance its clarity, and we look forward to addressing any further questions you may have.

---

### Official Review · Reviewer_R7PW · 2024-11-04

**Soundness:** 3
**Presentation:** 2
**Contribution:** 3
**Rating:** 5
**Confidence:** 4

**Summary:**

The paper introduces a new method - MIMOSA to interpret multimodal models.  It builds upon the success of using concept-based representations in unimodal settings. The authors propose that a shared concept representation across all modalities offers better insights into the model’s decision making process instead of having a concept representation for each modality. With the help of three different datasets - MNIST+, cdSprites+, CUB, the authors prove the accuracy of such a model. The authors also use individual modality decoders to extract information from the shared concept embedding.

**Strengths:**

1. Originality: The paper presents a new method to interpret multimodal models. The concept of using a single concept embedding with individual concept decoders is novel.

2. Quality: The paper provides a detailed analysis of the results, acknowledging that the model's task accuracy is suboptimal for certain datasets. The authors later also discuss why this is the case. Specifically, the sections “COMPARING SHARED AND SEPARATE EMBEDDINGS”, “EFFICACY OF INTERVENTION IN MULTIMODAL CONCEPT-BASED MODELS”, and “QUALITATIVE ANALYSIS OF REPRESENTATIVE CONCEPT EXAMPLES” are rife with information. These sections are also easy to follow as they have examples wherever needed. The example where the model interprets the 4 in an image as 3, shows the potential this method has.

3. Clarity: The paper is very well structured. The paper provides good context prior to the main methodology. Placing the related work section at the end helps maintain focus on this paper's contribution without disrupting the flow. The authors used good and relevant datasets for this work.

4. Significance: Multimodal interpretability has always been a challenging problem. There is a lot of research producing new black box models but very few talk about their interpretability. The paper correctly highlights that attention-based interpretability methods only show where the model is looking and not necessarily what the model is understanding. Knowing what the models are understanding can help us delve deeper into model performance. It would also help us identify biases in the models.
Overall, the paper shows a good promise towards interpreting multimodal models better.

**Weaknesses:**

1. Lack of clarity in the initial sections: The structure of the paper, particularly the ordering of the sections, is well-organized. The initial sections are very generic. Specifically, section 2 - Background and section 3 - Methodology. Concept-based representation is the key contribution of this paper. However, very little is mentioned about what that embedding could look like. The background section very briefly discusses concept based representations but those definitions are a little lacking and need to provide more insights. Understanding subsequent sections is challenging without prior knowledge of related work.. The section on multimodal representations does not offer much except the definitions of early and late fusion strategies in generic multimodal models. A lot of this section can be removed. Section 3 talks about the main methodology and is very brief and doesn’t provide enough information.

2. No Training Details: The paper talks about all of the models used for experiments but doesn’t provide any details about the training strategies used. Adding details about which models are pre-trained can help understand the paper a little better. Consider adding this information to the appendix of the paper. There are multiple networks in the paper potentially being trained on different objectives. Talking about the loss functions for each is helpful.

**Questions:**

1. Are all the encoder and decoder networks pre-trained? If not, what training procedure was used?
2. Could you provide a more detailed explanation of concept prediction and its significance within the context of this paper?
3. What training objectives are used for the decoder? How do we make sure that training the decoder doesn’t introduce any biases?
4. Why is the generalization capability of concept based models limited and how is concept embedding improving it?

---

> ### Author Response · Authors · 2024-11-20
> **Response to reviewer R7PW**
>
> Dear Reviewer R7PW,
>
> Thank you for your valuable feedback.
>
> Below, we provide responses to the specific concerns raised:
>
> ---
> #### 1. Clarity in initial sections
> Thank you for pointing out the issues on clarity. We will include the following clarifications:
> - Clarification on concept embeddings: Concept embeddings are fundamental components in concept-based models like MIMOSA, serving as vector representations of predefined, human-understandable concepts. These embeddings encapsulate the semantic and relational information of each concept, enabling the model to effectively interpret and reason about the input data across multiple modalities.
> - The discussion on multimodal representations will be streamlined to remove redundancy and focus on aspects directly relevant to our contributions, improving the flow and reducing confusion.
>
> #### 2. Training Details
> Thank you for pointing out the need for further details on the model structure.
> *MIMOSA* uses the following objectives:
>
> - **Concept Prediction Loss:** Binary Cross Entropy Loss for supervised concept predictions.
> - **Task Loss:** Cross-Entropy loss for the final classification task.
> - **Reconstruction Loss:** Modality-specific losses (e.g., MSE and SSIM for images, Cross-Entropy for text) ensure the alignment of shared concept embeddings with input modalities.
>
> The overall training objective combines these losses:
>
> $
> \text{L} = \lambda_\text{task} L_\text{task} + \lambda_\text{concept} L_\text{concept} + \lambda_\text{dec} L_\text{dec}
> $
>
> #### 3. Detailed Explanation of Concept Prediction and its Significance
>
> Concept prediction in MIMOSA involves assigning scores to predefined concepts based on the shared embedding space. These scores reflect the presence of concepts in the input data.
> This approach aligns with the principles of interpretable-by-design models, such as Concept Bottleneck Models (CBMs) (Koh et al., 2020) and Concept Embedding Models (CEMs) (Espinosa Zarlenga et al., 2022), where intermediate concepts are explicitly used to guide predictions.
>
> The significance of concept prediction lies in its ability to offer interpretable insights into the model's reasoning process, providing built-in interpretability and ensuring that the reasoning process is transparent and accessible without relying on post-hoc explanation methods.
> Concepts are designed to represent human-understandable units of information, allowing users to gain a deeper understanding of the factors influencing a model’s decision.
>
> Moreover, concept prediction enables interactive exploration through concept interventions. Users can modify the predicted concept scores to observe how changes in concepts affect the final task prediction.
>
> #### 4. Training Objectives for the Decoder and Bias Mitigation
> The decoders are trained to reconstruct the input data (images and text) from the shared concept embedding using reconstruction loss (e.g., Mean Squared Error or SSIM for images and Cross-Entropy for text).
>
> We did not specifically focus on bias mitigation in this work, as our primary objective was to achieve accurate reconstructions and robust concept-based interpretability. However, our model might help in the detection of these biases by means of concept visualizations.
>
> Indeed, while standard CBMs only allow us to check the concept predictions, our model also enables visualization of the concept representations, which can highlight the presence of biases. As an example of bias that we found, on the MNIST+ dataset, on input digits particularly thin, the model was more prone to predict them as 1 or 7, while instead, they were 8 or 9.
>
> We will better describe this in the paper, acknowledging the importance of identifying and addressing potential biases.
>
> #### 5. Generalization Capability of Concept-Based Models and Improvement through Concept Embeddings
> The architecture of Concept Bottleneck Models (CBMs) relies on a concept layer where each concept is represented by a single neuron.
> While this design facilitates interpretability, it imposes a bottleneck on the model's representation capacity, often limiting its generalization performance, as highlighted in prior research [Koh et al., 2020].
>
> In contrast, Concept Embedding Models (CEMs) [Espinosa Zarlenga M. et al., 2022] leverage richer concept representations by using vectorial embeddings, which provide more expressive and flexible representations of concepts.
>
> MIMOSA builds upon CEM by integrating not just richer concept embeddings but also multimodal information from both visual and textual modalities. This multimodal approach enhances the model's generalization capabilities while maintaining its interpretability.
>
> ---
>
> Thank you again for your detailed feedback. We believe that these updates will strengthen our paper and enhance its clarity, and we look forward to addressing any further questions you may have.

---

> > ### Comment · Reviewer_R7PW · 2024-11-25
> >
> > Thank you for your explanation. However, I still have some comments:
> >
> > 1. The revisions to the paper seem minimal. Crucial training details are still missing from both the main paper and the appendix. In this research area, comprehensive training details are essential for reproducibility and evaluation.
> > 2. Your response doesn't clarify the paper's novel contributions. Much of the work appears to replicate existing unimodal approaches. Please elaborate on the unique aspects of your multimodal methodology.
> > 3. Bias mitigation in decoder networks is very important as these networks are offering insights into the actual model. If the decoder is biased, how would you know if the output visualization is because of a main model bias or a decoder bias?
> > 4. While I agree that CEMs offer greater generalizability than CBMs, my concern lies with the inherent limitations of CEMs, regardless of modality. The transition from unimodal to multimodal contexts should primarily expand the concept space without impacting the generalization capabilities of the CEM itself. Could you elaborate on the potential limitations in generalizability within your multimodal CEM framework?

---

### Note · Authors · 2024-12-04

**Comment:**

We thank the reviewers for their valuable feedback. We plan to refine our work based on their suggestions and submit it to a future conference.

**Withdrawal Confirmation:**

I have read and agree with the venue's withdrawal policy on behalf of myself and my co-authors.